



# Technical note: Controversial aspects of the direct vapor equilibration method for stable isotope analysis (δ¹⁸O, δ²H) of matrix-bound water: Unifying protocols through empirical and mathematical scrutiny

Benjamin Gralher, Barbara Herbstritt, Markus Weiler

Chair of Hydrology, University of Freiburg, Freiburg 79098, Germany

*Correspondence to*: Benjamin Gralher (benjamin.gralher@googlemail.com)

**Abstract.** The direct vapor equilibration laser spectrometry (DVE-LS) method has been developed for obtaining matrix-bound water stable isotope data in soils, the critical zone and bedrock, deriving therefrom subsurface water flow and transport processes and, ultimately, characterising e.g. groundwater recharge and vulnerability. Recently, DVE-LS has been increasingly adopted due to its possible high sample throughput, relative simplicity and cost-efficiency. However, this has come at the cost of a non-unified standard operation protocol (SOP) and several contradictory suggestions regarding protocol details do exist which have not been resolved to date. Particularly, sample container material and equilibration times have not yet been agreed upon. Beside practical constraints, this often limits DVE-LS applicability to interpreting relative isotope dynamics instead of absolute values. It also prevents data comparability among studies or laboratories and several previous comparisons of DVE-LS with other, more traditional approaches of water extraction and subsequent stable isotope analysis yielded significant discrepancies for various sample matrices and physical states. In a series of empirical tests, we scrutinized the controversial DVE-LS protocol details. Specifically, we tested ten different easily available and cost-efficient inflatable bags previously employed or potentially suitable for DVE-LS sample collection and equilibration. In storage tests similar to the DVE-LS equilibration process but lasting several weeks, we quickly found heat-sealed bags made of laminated Aluminum (Al) sheets to be superior by several orders of magnitude over more frequently used freezer bags in terms of evaporation-safety and accompanying adverse isotope effects. For the first time, Al-laminated bags allow the applied equilibration time to be adapted exclusively to sample requirements instead of accepting reduced data quality in a trade-off with material shortcomings. Based on detailed physical considerations, we further describe how to calculate the minimum available container headspace and sample-contained liquid water volume and how their ratio affects analytical precision and accuracy. We are confident, that these guidelines will expand DVE-LS applicability and improve data quality and comparability among studies and laboratories by contributing to a more unified, physically well-founded SOP based on more appropriate components.



## 1 Introduction

The direct vapor equilibration laser spectrometry (DVE-LS) method first published by Wassenaar et al. (2008) has facilitated a way for fairly convenient, high-throughput stable isotope analysis of water bound to the soil matrix, rocks or plant tissue. The method bypasses many of the previously necessary, laborious sample preparation steps. At the same time, it increases the number of samples that can be processed per day. It employs inflatable sample containers into which evaporation-susceptible soil, rock or plant samples of interest are quickly collected. Following sample collection, the containers are

commonly inflated with a dry inflation atmosphere and sealed. Then they are left for isothermal isotope equilibration between the matrix-bound liquid water reservoir of interest and the container headspace atmosphere vapor prior to the direct, yet non-automated analysis of the water vapor via laser-based isotope spectrometry. A schematic drawing of the DVE-LS methodic steps is shown in Figure 1. Co-measured calibration standards are referenced to the VSMOW-SLAP scale (Craig, 1961) and prepared accordingly, following the Principle of Identical Treatment (PIT) (Werner and Brand, 2001). They allow

for straightforward calculation of sample liquid water stable isotope signatures from the standards' known liquid water isotope signatures and raw headspace water vapor isotope readings of standards and samples.

The growing distribution of laser-based water stable isotope analyzers in recent years and the DVE-LS method's relative simplicity manifested in fairly little sample preparation workload, low-cost consumables and omission of sophisticated water extraction lines and analyzer peripherals enabled its rapid, wide-spread adoption. It has now been employed to investigate a

long list of processes and phenomena in hydrology, ecohydrology, pedology, hydrogeology and related disciplines spanning the entire plant-soil-groundwater continuum in various climates. Unlike e.g. suction cups or mechanical squeezing, DVE-LS is assumed to provide isotope data that are not tension-specific but represent the bulk water of a given sample (Sprenger et al., 2015a). On the hillslope scale, the DVE-LS method has been used to reveal present and past subsurface water flow paths in the unsaturated and saturated zone of humid (Garvelmann et al., 2012) or alpine regions (Mueller et al., 2014). On a

similar scale, it has been used to obtain high resolution water isotope depth profiles for the investigation of spatial and temporal dynamics of water flow and solute transport in a heterogeneous glacial till (Stumpp and Hendry, 2012). Sprenger et al. (2015b) used it to test and compare different modeling strategies to determine soil water flow and solute transport parameters. On the regional scale, it was employed to quantify the spatiotemporal variability of tree water uptake (Bertrand et al, 2014), to evaluate aquifer recharge and vulnerability in an alluvial lowland (Filippini et al., 2015), to assess snowmelt-

dominated groundwater recharge in a northern region (Chesnaux and Stumpp, 2018; Boumaiza et al, 2020), and to feed a groundwater recharge model for ungauged watersheds (Mattei et al., 2020). With the help of DVE-LS data from the shallow subsurface, the impacts of the 2018 drought in Central Europe and its recovery on subsurface water stress, water ages and ecohydrologic fluxes were understood and simulated (Kleine et al., 2020; Smith el al., 2020). In the deep saturated zone, the DVE-LS method helped to interpret high-resolution depth profiles and thus retrace paleogroundwater flow and long-term

transport processes in aquitards (e.g. Hendry and Wassenaar, 2009, 2011; Hendry et al., 2011a, 2013; Harrington et al.,





2013). In all these examples DVE-LS analyses were performed on soil or rock samples. Although generally conceivable (Millar et al., 2018, 2019), we are not aware of any field study employing DVE-LS on plant samples.

In principle, the DVE-LS method rests upon analyzing a corresponding vapor phase instead of the liquid water reservoir of interest itself. Meanwhile, this working principle has been transferred even to continuous, minimally-invasive in situ
approaches of stable isotope analysis of water that is either freely flowing (Munksgaard et al., 2011; Koehler and Wassenaar, 2011; Herbstritt et al., 2012) or bound to the matrix of soils (Rothfuss et al., 2013; Volkmann and Weiler, 2014) or plant xylems (Volkmann et al., 2016). The calibration of so-obtained isotope data has also been aided in some cases via DVE-LS analyses of carefully prepared standards (e.g. Oerter et al., 2016).

The DVE-LS method employs laser-based isotope analysis. It does therefore not come completely without complications.
Generally, it has been demonstrated that laser-based stable isotope analyzers are susceptible to the influence of gaseous contaminants like alcohols (Brand et al., 2009; Martín-Gómez et al., 2015) which may be emitted from plant samples, or $H_2S$ (Malowany et al., 2015) or methane (Hendry et al., 2011b) which may appear in anoxic or contaminated sites. Accordingly, this is also relevant for DVE-LS analyses performed on samples from such origins. Hendry et al. (2011b) described and tested a correction algorithm applicable for analyzers based on cavity ring-down spectrometry (CRDS) that are exposed to
naturally occurring methane levels. For samples contaminated with methanol or ethanol, Martín-Gómez et al. (2015) compared a self-developed post-processing software with an on-line oxidation oven (Micro Combustion Module, Picarro) physically implemented into the measurement process of a Picarro L2120-*i* and were able to correct or remove considerable levels of these contaminants. However, they did not test their setup for DVE-LS analyses. The impact of changing background gas matrices, which may happen e.g. due to ongoing microbial activity in natural soil samples, has been
investigated by Gralher et al. (2016). They also presented a post-correction scheme based on an analyzer-recorded spectral variable and measurement iterations of potentially affected DVE-LS samples (Gralher et al., 2018).

Overall, the DVE-LS method has considerably simplified matrix-bound water stable isotope analysis. However, it is not yet perfect and several studies have aimed at specifically testing and/or improving accuracy, precision and/or the general applicability of the protocol originally described by Wassenaar et al. (2008), e.g. by comparison with other methods of
matrix-bound water stable isotope analysis. Hendry et al. (2015) compared DVE-LS results against isotope analyses of water obtained from piezometers and mechanical squeezing of geologic core samples. They suggested spiking the employed drilling fluid with $^2H$ to detect contamination of original pore water which they observed e.g. in samples from saturated, highly permeable geologic media. They also tested different sample storage containers and favored Ziploc® freezer bags, which when doubled they found to reliably hold sample water and prevent significant evaporitic enrichment of heavy
isotopes for up to ten days. Comparing the DVE-LS method with analyses of liquid water squeezed from low-permeability samples, Nakata et al. (2018) found the former to represent water from open pores only. Millar et al. (2018) analyzed plant samples from a controlled environment in a direct comparison of the DVE-LS method against five quantitative water extraction methods. They found the former to be superior in terms of limited co-extraction of volatile organic compounds




(VOCs), rapid sample throughput, and near-instantaneously returned stable isotope results. They reported, however, that the

DVE-LS method systematically yielded water stable isotope signatures somewhat enriched in $^2$H and $^{18}$O content.

Mattei et al. (2019) scrutinized the DVE-LS method on an analyzer employing off-axis integrated cavity output spectrometry (OA-ICOS). For the projected calibration of samples, they investigated Fold-A-Carrier Reliance™ bags (Reliance Products, Winnipeg, Canada) of 20 L volume filled with 20 mL water aliquots. They found the bags to retain 99% of the injected water over the course of 30 days. They highlighted the possibility of many measurement iterations at the cost, however, of a high

consumption of standard water volume. They also addressed vapor concentration effects on their instrument which caused high variabilities in isotope readings between different combinations of water vaporizing methods and modes of analyzer operation. Testing their approach on oven-dried-and-rewetted soil aliquots, they found isotope readings to reach plateaus after six days of equilibration. Such soil aliquots were also used by Wang et al. (2019) who tested different isotope data correction strategies including one incorporating soil physical variables, namely relative clay and water content. They

suggested that the correction strategy should be adapted to the research focus of isotope assays (e.g. groundwater vs. root water uptake). Using Ziploc® freezer bags, they defined the optimum equilibration time to be 12-24 h and argued that longer equilibration times should be avoided due to the onset of evaporitic enrichment of heavy isotopes afterwards.

Ziploc® bags as employed in the original DVE-LS study (Wassenaar et al., 2008) are used by many research groups. They are re-sealable, inflatable, considered sufficiently leak-tight, and collapsible as demanded. Eventually tested alternatives

fulfilling the same criteria have been found to be prohibitive for large-scale applications. However, a Ziploc® bag's diffusional barrier is clearly not absolute and thus restricts proposed maximum equilibration times. It stands out that suggested equilibration times vary considerably. They range from 24 h (Wang et al., 2019) to 10 days (Hendry et al., 2015). Notably, they are consistently substantiated by water loss and accompanying adverse effects on isotope data upon exceedance. This may be indicative for differences in the bags' production processes and/or their storage conditions during

the respective investigations. Either way, this is not satisfying as it makes so-obtained suggestions not generally transferable between laboratories. Also, ideal equilibration times should not be defined by the containers' shortcomings but exclusively by the samples' properties and best possible data quality. This holds also for extreme soil physical settings regarding e.g. sample permeability or size.

Despite the large number of studies aiming at improving the DVE-LS method, only Hendry et al. (2015) combined

headspace isotope analyses and weight loss observations on identical samples. And to date, no material has been scrutinized that is suitable for DVE-LS sample bags and allows for storage times which are not restricted by the bags' shortcomings while at the same time still coming at reasonable per-unit-costs and additionally fulfilling all criteria listed above. Such bags would also simplify the sample handling process prior to equilibration. They would dispense with additional workload and potential pitfalls in cases of restricted or delayed laboratory access. In such a case, Ziploc® bags require extra measures like

e.g. evaporation-susceptible sample transfer from evaporation-safe containers such as glass jars (Mattei et al., 2019) or precautionary deposition in coolers (Wassenaar et al., 2008). Evaporation-proof sample bags would also allow for the calculation of minimum sample sizes based exclusively on physical requirements and thus likely expand the applicability of



the DVE-LS method regarding the range of potential sample size, matrices and physical states. Finally, they would expand the DVE-LS method's reliability in terms of interpreting absolute isotope values instead of being limited to relative

dynamics in the case of e.g. Deuterium-labelled samples quickly inducing adverse isotope effects due to extraordinarily high vapor pressure gradients across container walls on the isotopologue level. In summary, no unified standard operation protocol (SOP) exists to date. Unfortunately, this bears the risk that unsuitable protocol details are applied in inappropriate cases.

Therefore, the aim of this study was to further improve the trustworthiness of DVE-LS data and to allow comparability

across laboratories by finding improved components and contributing to a more unified SOP. Specifically, we wanted to experimentally identify better, yet affordable materials for DVE-LS sample storage containers. Then, we determined minimum and maximum storage and equilibrium times that are not dictated by gradual water loss and evaporitic enrichment of heavy isotopes. We simulated potential evaporative water losses with a Rayleigh-type approach. Finally, we aimed at mathematically assessing reasonable container and sample sizes which we deem necessary for obtaining accurate and precise

DVE-LS-derived isotope data of matrix-bound water reservoirs.

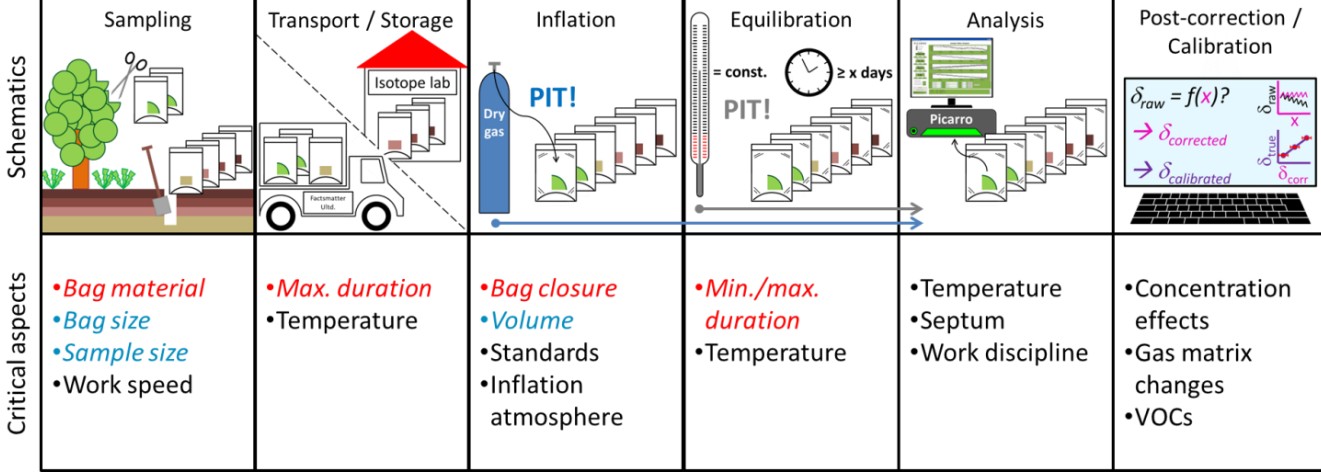

**Figure 1: Schematics and critical aspects of the DVE-LS methodic steps. Aspects quantitatively investigated in this study are highlighted in red (empirical) and blue (mathematical).**

## 2 Method

### 2.1 Empirical observations

### 2.1.1 Material selection

In the first part of our study, we intended to get an overview of different bags that are potentially suitable for DVE-LS. Consequently, we were looking for materials that can be used to build airtight, inflatable, collapsible, and resealable bags as originally demanded by Wassenaar et al. (2008). Our investigations focused on food storage products due to their wide-




spread use and resulting easy availability and relative inexpensiveness. We finally obtained ten different bags from commercial sources including those previously used for DVE-LS applications and two custom-made bags from local fish and meat vendors, originally intended for keeping their products isolated and odorless after hand-out to customers. With these bags we closely simulated the originally proposed DVE-LS protocol. Specifically, all bags were filled with 233 g to 380 g of field-moist soil, inflated and closed by means of an integrated zip closure where available, via heat-sealing if

accordingly designed, or with Teflon (PTFE) sealing tape (Petri-Seal$^{TM}$, Sigma-Aldrich) otherwise. The bags were left on the laboratory bench in a temperature-controlled environment (20°C ± 1°C) exposed to ambient air (RH: 11.8% – 83.1%, mean: 42.3% ± 11.6%) and occasionally weighed (PT3100, Sartorius, Göttingen, Germany, www.sartorius.com, resolution: 0.1 g) over the course of up to 71 days. This part of the study was conducted on unique items and served as a pre-test to the actual assessment of isotope effects potentially complicating DVE-LS analyses. A list of characteristics of the bags used in this part

of our investigations can be found in Table 1.

**Table 1: Characteristics of the different bags tested in the first part of this study.**

| Bag ID | Commercial product name & manufacturer | Material type | Material strength (µm) | Closing mechanism | Water vapor permeability (manufacturer information) | Dimensions W × H (cm × cm) | Approx. volume (L) |
|---|---|---|---|---|---|---|---|
| G&G | Gut & Günstig Gefrierbeutel[ede] | LDPE | 70 | zip closure | 1 g/(m² d)* | 17.5 × 20.5 | 1 |
| FF1 | fish packaging foil[MIG] | Al-steamed plastic | total: 60 Al: N/A | heat-sealed | N/A | 15 × 30 | 2 |
| toppits | Toppits® Gefrierbeutel[Cof] | LDPE | 100 | double Ziploc® | 1 g/(m² d)* | 17.5 × 20.5 | 1 |
| SBz_t | CB400-524VtZ[WP] | PET-PE-LDPE | total: 130 | zip closure + PTFE tape | 0.4 g/(100 in² d)** | 20 × 27 | 2.4 |
| FF2 | fish packaging foil[WP] | Al-/ plastic-coated paper | total: 95 Al: N/A | heat-sealed | N/A | 17 × 30 | 2.5 |
| s_Al3z | CB400-311BRZ[WP] | PET-Al-LDPE | total: 127 Al: 7.1 | zip closure | < 0.02 g/(100 in² d)** | 13.5 × 18.5 | 0.8 |
| Al3z | CB400-420GBZ[WP] | PET-Al-LDPE | total: 127 Al: 7.1 | zip closure | < 0.02 g/(100 in² d)** | 14.5 × 24 | 1.2 |
| h_Al3z | CB400-528N[WP] | PET-Al-LDPE | total: 127 Al: 7.1 | zip closure | < 0.02 g/(100 in² d)** | 20 × 26 | 2.4 |
| Al3_t | CB300-510N[WP] | PET-Al-LDPE | total: 127 Al: 7.1 | PTFE tape | < 0.02 g/(100 in² d)** | 20 × 26 | 2.4 |
| Al3z_hs | CB400-420BRZ[WP] | PET-Al-LDPE | total: 127 Al: 7.1 | zip closure + heat-sealed | < 0.02 g/(100 in² d)** | 14.5 × 24 | 1.2 |

[ede] = edeka, Germany (vendor)
[MIG] = MIGROS, Switzerland (vendor)
[Cof] = Cofresco, Minden, Germany
[WP] = Weber Packaging, Güglingen, Germany


* = @ 85% RH, 23°C (URL1)
** = @ 90% RH, 40°C (pers. comm. PACIFIC BAG INC, Woodinville, WA, USA)



### 2.1.2 Weight losses and stable isotope effects

In the second part of the study we focused on quantitatively assessing the effects of selected storage bags on DVE-LS-based stable isotope analysis of matrix-bound water. For this purpose, we reduced the number of different bags but increased the
number of replicates. We selected bag candidates that spanned the largest part of the weight losses observed in the first part of the study. Additional, rather pragmatic aspects of this selection process were the bags' ruggedness and expected ease of handling during projected, time-critical collection of large numbers of evaporation-susceptible soil, rock or plant samples in the field. In total, 21 replicates of each of these bag candidates were then equipped on one side with custom-made septa of silicone blots or adhesive tape. This time we differed from the original DVE-LS protocol by omitting the soil. Instead, all
bags were filled with 5 mL of isotopically identical pure water aliquots, inflated with dry air, sealed immediately thereafter and weighed (PCB2500-2, Kern & Sohn, Balingen, Germany, www.kern-sohn.com, resolution: 0.01 g). Again, the bags were left in a temperature-controlled environment and exposed to ambient air. After 1, 2, 5, 9, 14, 21, and 28 days successive subsets of three replicates of all bag versions were weighed again and their headspace water vapor stable isotope signatures ($\delta^{18}O$ and $\delta^2H$) were determined.
Isotope analysis was facilitated by puncturing the bags through the previously applied septa with a hollow needle directly connected via 1/8" Teflon (PFA) tubing to the sample inlet port of the cavity ring-down isotope analyzer (L2120-$i$, Picarro Inc., Santa Clara, CA, USA, www.picarro.com). On each day of analysis, reference standards were co-measured to account for potential instrument drift or unintended fluctuations of laboratory air temperature. Each time, these standards had been freshly prepared two days in advance. For this purpose, 5 mL of identical water aliquots had been filled into the bags with
the lowest water loss rate observed in the first part of this study. Otherwise, this preparation followed the Principle of Identical Treatment (PIT) between samples and standards. For each day, we calculated and report here the differences in raw isotope readings between the sample triplicates and respective standards.

### 2.2 Data analysis

We assumed that potential weight losses of the bags would occur solely due to evaporation and diffusion of water vapor out
of partly gas-permeable bags, and isotope data of the liquid water reservoir would then follow a Rayleigh-type evolution (Lord Rayleigh, 1902). Thus, liquid water isotope signatures were calculated by using raw isotope readings of the bags' sampled headspace vapor and a linear relationship between the standards' headspace readings and referenced liquid water isotope values assuming a slope of 1 which had been repeatedly confirmed in liquid water isotope analyses on the same instrument. Then, for oxygen ($^{18}O/^{16}O$) and hydrogen ($^2H/^1H$) the isotope ratio $R$ was simulated with the following approach

$$R = R_0 * f^{\alpha-1} \tag{1}$$

where subscript $0$ refers to the start of the observations, $f$ is the remaining fraction of the water reservoir at the respective time of observation, and $\alpha$ is the isotopic fractionation factor between the liquid water reservoir and the evolving vapor. Further, it holds





$$R = R_{std} * \left(1 + \frac{\delta_{sam}}{1000‰}\right) \tag{2}$$

where $\delta$ denotes the isotope signature in delta notation and the subscript *std* in this case refers to the respective international standard for oxygen and hydrogen stable isotope ratios, VSMOW (Craig, 1961).

We calculated the minimum sample bag headspace volume $V_{hsp}$ necessary for precise replicate analysis of matrix-bound water isotopes via DVE-LS using equation (3), where *n* is the number of desired, safely possible measurement iterations and also accounts for occasionally necessary prolonged analyses, *q* is the analyzer-demanded gas flow rate (in mL/min) and *t* is

the time period (in min) usually necessary for reaching a sufficiently long plateau during numerous DVE-LS analyses previously conducted in our laboratory:

$$V_{hsp} = n * q * t \tag{3}$$

We calculated the minimum necessary liquid water reservoir contained in the sample of interest $V_{H2O,sam}$ using equation (4), where $\varepsilon(T_{air})$ is the isotope separation at equilibration temperature (in ‰$_{VSMOW}$), $V_{H2O,eq}$ is the liquid water equivalent of the

water present in the vapor phase of the bag (e.g. in m³) and $\Delta\delta_{acc}$ is the isotope-specific accepted measurement uncertainty (in ‰$_{VSMOW}$) that must not be exceeded systematically.

$$V_{H2O,sam} = \frac{\varepsilon(T_{air}) * V_{H2O,eq}}{\Delta\delta_{acc}} \tag{4}$$

The isotope separation was calculated using equation (5) where α is the temperature-dependent isotope equilibrium fractionation factor between liquid water and a corresponding vapor phase (Majoube, 1971).

$$\varepsilon(T_{air}) \approx (\alpha(T_{air}) - 1) * 1000‰ \tag{5}$$

The liquid water equivalent was calculated using equation (6) where $T_{K,air}$ is the equilibration temperature (in K), $V_{bag}$ is the bag headspace volume (in m³), *R* is the gas constant, $p_{air}$ is air pressure, $M_{H2O}$ is the molar mass of water, and $\rho_{H2O}$ is the density of liquid water.

$$V_{H2O,eq} = \frac{V_{bag}}{\frac{R*T_{k,air}}{p_{air}}} * \frac{E_{H2O}(T_{air})}{p_{air}} * \frac{M_{H2O}}{\rho_{H2O}} \tag{6}$$

The first term on the right side of this equation accounts for the ratio of the bag volume and the volume one mole of gas occupies under given conditions, the second term expresses the share that water vapor has of total molecules present in the gas phase and the third term converts the previous ones from a mole number into a volume of liquid water. With $R =$ 8.314 J/(mol K), $\rho_{H2O}$ = 1000 kg/m³, $M_{H2O}$ = 0.018 kg/mol and canceling out the air pressure $p_{air}$, equation (6) simplifies to:

$$V_{H2O,eq} = \frac{V_{bag} * E_{H2O}(T_{air})}{T_{K,air}} * 2.165 * 10^{-6} \, K/Pa \tag{7}$$

$E(T_{air})$ is saturation vapor pressure (in Pa) as a function of air temperature (Foken, 2008). It is calculated with equation (8) where $T_{air}$ is air temperature (this time in °C).

$$E_{H2O} = 611.2 * e^{\frac{17.62 * T_{air}}{243.12 + T_{air}}} \tag{8}$$

Equation (9) is somewhat similar to equation (4). It is based on closed-system assumptions inside a sample container and the fact that the residual liquid water isotopic composition is systematically shifted towards "heavier" values when a significant





fraction thereof (1-*f*) saturates an initially dry atmosphere to achieve isotope equilibrium (see e.g. line D of Fig. 2 in Gat, 1996). Assuming a linear relationship between the remaining water fraction *f* and changes of its isotopic composition and applying intercept theorem and mass balance considerations we obtain:

$$\frac{\Delta\delta_{cs}}{\varepsilon_{eq}} = \frac{V_{H2O,eva}}{V_{H_2O,sam}} = (1 - f) \qquad (9)$$

where $\Delta\delta_{cs}$ is the systematic shift of both vapor and liquid water stope signatures caused by equilibrium fractionation in a

closed system, $\varepsilon_{eq}$ is the equilibrium isotope separation again (Eq. 5), $V_{H2O,eq}$ is the evaporated water volume (Eq. 7), and $V_{H2O,sam}$ is the total liquid water volume initially present in the sample.

Ratios of mean isotope enrichment rates were calculated as estimates of the slopes of so-called evaporation lines that water stable isotope data plot on in dual isotope space when affected by gradual evaporitic enrichment of heavy isotopes. We compared these to the ratio of deviations from unity of the model-derived isotope fractionation factors α (Eq. 1).

Individually, these deviations yield the respective isotope separations (Eq. 5).

## 3 Results

### 3.1 Empirical observations

#### 3.1.1 Material selection

The average area-normalized weight loss rates of the ten tested bags varied by three orders of magnitude, ranging from

0.006 g/(m² day) to 1.415 g/(m² day). They were highest for the transparent low density polyethylene (LDPE) bag of low strength (G&G) and lowest for the heat-sealed bag that included one layer of aluminum (Al) foil (Al3z_hs). For non-metalized bags the weight loss rates were in the opposite order of their wall strengths. For metalized bags they were highest where the Al layer had been applied by a steaming process and lower in cases of laminated Al foil. For the latter, they were highest for zip-closed-only bags and lowest for the additionally heat-sealed bag, with the PTFE-taped bag (Al3_t) plotting in

between, close to Al3z_hs. For the three Al-laminated bags with identical closures, differing only in their sizes, the largest one (h_Al3z) had a slightly lower weight loss rate than the other two (Al3z and s_Al3z) which were almost identical. The different lengths of the time series occurred because only few bag types were available at the beginning of the observations and additional specimen (+ heat-sealing pliers) were found later and included. Observations were terminated when clear trends had become visible for all bags under investigation. Time series of weight losses are displayed with a synchronous

start for better comparability of trends (Fig. 2). For all time series clear linear relationships were found with coefficients of determination (R²) higher than 0.98 for those exceeding absolute weight losses of 0.2 g.





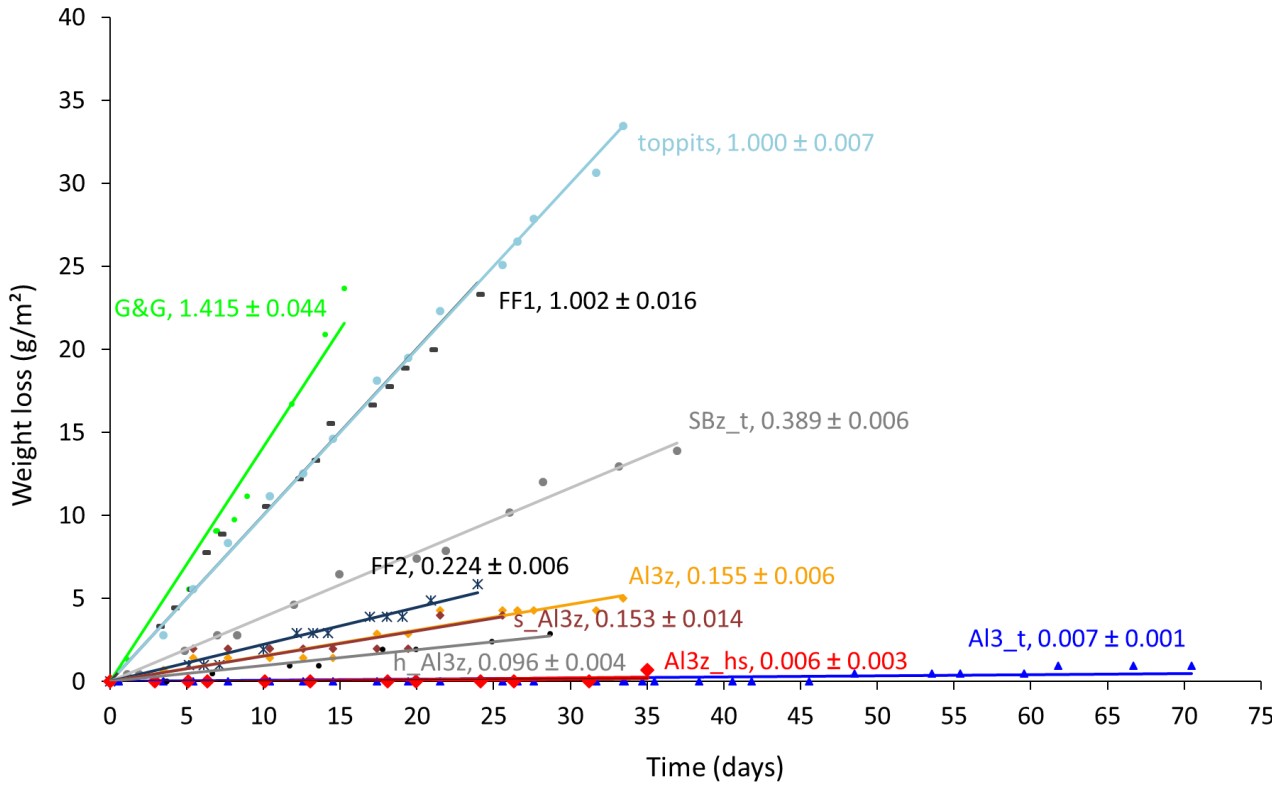

**Figure 2: Time series of weight losses normalized to the surface areas of ten different bags filled with moist soil. Numbers in line labels indicate average weight loss rates and uncertainties in g/(m² day).**

**3.1.2 Weight losses and stable isotope effects**

For further assessment we selected the Toppits® Ziploc® freezer bags and the Al-laminated bags with zip closure and a volume of 1.2 L. Both bag types are available at reasonable costs: the former are sold by regular household supply stores (€ 0.14/unit) while the latter can be obtained from a specialty packaging wholesaler (see Tab. 1) (€ 0.65/unit). Toppits® freezer bags were used as standalone (single layer, "toppits") and bag-in-bag (double layer, "toppits_double") version. Al-

laminated bags were used as zip-closed-only ("Al3z") and as zip-closed and additionally heat-sealed ("Al3z_hs") version. This time, the area-normalized weight loss rate of toppits and Al3z bags had decreased by 16.3% and 23.2%, respectively, compared to the previous results while it was – although on a low level – higher for Al3z_hs bags. Nonetheless, weight loss patterns were congruent in both parts of the study. Again, highest weight loss rates were observed for transparent, non-metalized bags having the lowest barrier strengths (toppits) and lowest weight loss rates were observed for Al-laminated,

heat-sealed bags (Al3z_hs). On the final day of our experiment, mean weight losses reached 1.71 g and 0.90 g for single and double layer Toppits® bags, respectively. In the same order, these weight losses represent 34.1% and 17.9% of the weight of



the water initially filled into the bags. For Al3z and Al3z_hs bags, we observed final average weight losses of 0.23 g and 0.06 g which translate to 4.5% and 1.1%, respectively. Averages and uncertainties of normalized weight loss rates for all bag versions can be found in Table 2. For transparent and Al-laminated bags mean standard devations of triplicates were lower

for the double-walled and more thoroughly closed version, respectively, and generally inconsistent over time.

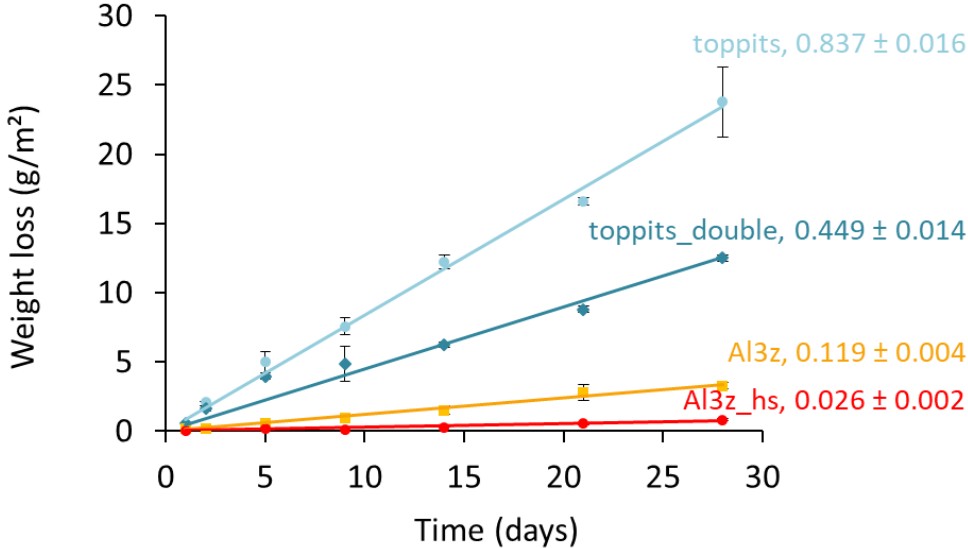

**Figure 3: Time series of average weight losses normalized to the surface areas of single layer (light blue circles) and double layer (blue diamonds) Toppits® freezer bags as well as zip-closed-only (orange squares) and additionally heat-sealed (red circles) Al-laminated bags. Numbers in line labels indicate average weight loss rates and uncertainties in g/(m² day).**


Unlike weight loss data, temporal changes of isotope readings were not normalized to the bags' surface areas in order to enable direct comparison with generally accepted measurement uncertainties. Over the course of 28 days headspace vapor isotope readings changed steadily for three of the four bag versions. At the end of the observation period they deviated from initial readings on average by +10.23‰, +4.74‰, and +1.37‰ for $\delta^{18}O$ and +37.34‰, +20.00‰, and +2.78‰ for $\delta^2H$

for toppits, toppits_double and Al3z bags, respectively. In the same order, changes of $\delta^{18}O$ exceeded the standard deviation derived from replicates of the co-measured standards after 2, 5, and 21 days and changes of $\delta^2H$ crossed this margin after 2, 2, and 28 days. For heat-sealed Al-laminated bags (Al3z_hs), no trend exceeding these standard deviations within the observation period was found for both isotope signatures under investigation. Due to "noisiness" on day 1, linear regression models were applied starting day 2 (Fig. 4). Their slopes decreased with increasing barrier strengths in the case of Toppits®

bags and more thorough closures in the case of Al-laminated bags. This pattern is consistent with water loss characteristics (Fig. 3). For all bag versions investigated, the ratio of isotope enrichment rates, which yields the slope of a so-called evaporation line in dual isotope space, is consistently lower than 8 (which would have indicated isotope equilibrium). It is





fairly equal for both freezer bag versions, considerably lower for AL3z bags and even negative for Al3z_hs bags. For the

latter the respective underlying average isotope enrichment rates are exceeded by their uncertainties. Absolute numbers and

uncertainties of change rates of isotope readings can be found in Table 2. With only few exceptions (toppits: day 5 and 28),

standard deviations of triplicate isotope analyses were smaller than the observed drift standard measurement uncertainty

(0.6‰ for $\delta^{18}O$, 2.14‰ for $\delta^2H$).

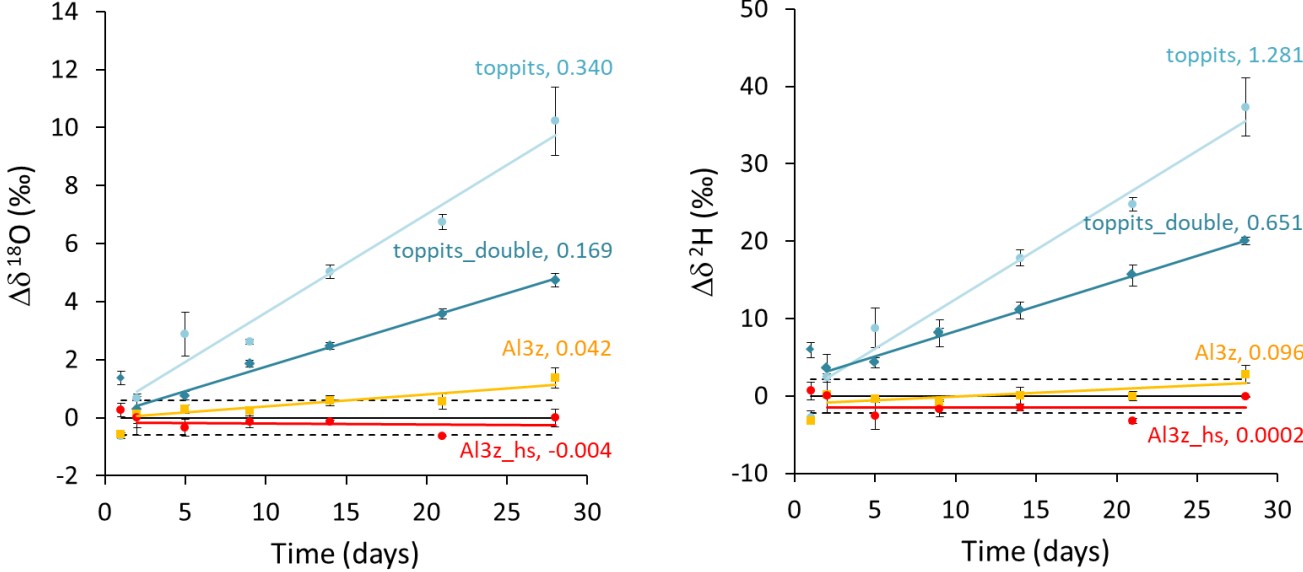

**Figure 4: Time series of drift-corrected changes in headspace water vapor stable isotope readings observed in triplicates of single**

**(light blue circles) and double layer (blue diamonds) Toppits® freezer bags, of zip-closed-only (orange squares) and additionally**

**heat-sealed (red circles) Al-laminated bags. Numbers in line labels indicate average changes of isotope readings in ‰/day after day**

**1. Horizontal lines represent baselines (solid lines) and reference water vapor standard deviations of ±0.60‰ and ±2.14‰ for $\delta^{18}O$**

**and $\delta^2H$, respectively (dashed lines).**

### 3.2 Data analysis

The outcome of the Rayleigh-type simulation (Eq. 1) of the calibrated liquid water isotope signatures obtained from the

headspace vapor observations can be seen in Figure 5. For all bag types and both isotope ratios investigated the observed

isotope evolutions do not exceed the quasi-linear part of a typical Rayleigh curve. For the Toppits® freezer bags isotope data

plot along clear paths. For the Al-laminated bags, isotope data and remaining fraction data calculated from weight

observations plot within very narrow ranges not displaying any distinct correlations or trends. The respective isotope

fractionation factors were determined by minimizing the root mean squared error (RMSE) between observed and simulated

isotope data using Equation 1 and the SOLVER function of the Microsoft® EXCEL software package. In the case of toppits

bags, deviations from unity of the so-obtained fractionation factors (compare Eq. 5) were about twice as high as the ones of





toppits_double bags for both isotope ratios. The deviations from unity were highest for Al-laminated bags with absolute
values being inverted for Al3z_hs bags. Numerical values of all model-derived fractionation factors can be found in Table 2.

RMSE values as functions of fractionation factors were calculated as a measure of parameter sensitivity and are shown in
Figure 6. Overall, we arbitrarily varied fractionation factors by ±1 relative to their respective RMSE-optimized values which
extends the range we consider physically possible. Over the entire range investigated, RMSE values displayed only the one
minimum presented here for each bag type and isotope ratio. Minimum RMSE values as well as relative changes thereof
were lowest for Al-laminated bags and highest for freezer bags for both isotope fractionation factors determined. The ratios

of observed mean enrichment rates and the ratios of deviation from unity of the model-derived fractionation factors were in
very good agreement in the case of toppits bags and differed most in the case of heat-sealed Al-laminated bags (Tab. 2).



**Figure 5: Observations (diamonds, "obs") and Rayleigh-type simulations (red lines, "sim") of liquid water $\delta^{18}$O (left column) and $\delta^2$H data (right column) obtained from single (top row) and double layer (second row) Toppits® freezer bags and zip-closed-only (third row) and additionally heat-sealed (bottom row) Al-laminated bags as functions of the respective residual water fractions $f$.**



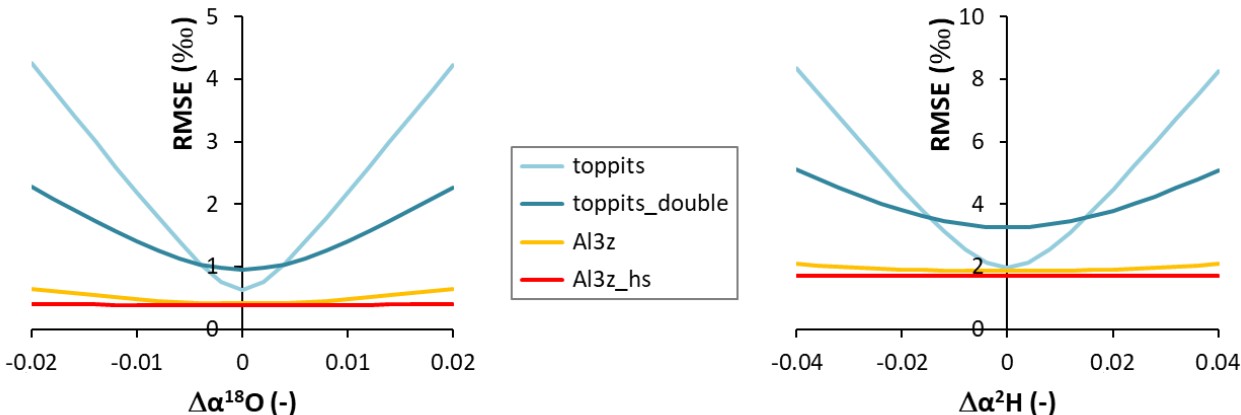

**Figure 6: Absolute model RMSE values as functions of the absolute deviations from the RMSE-optimized values of $\alpha^{18}O$ (left) and $\alpha^2H$ (right).**


**Table 2: Characteristics of changes in weight and isotope readings and model-derived isotope fractionation factors.**

| Bag ID | Weight loss rate (g/(m² day)) | $\delta^{18}O$ enrichment rate (‰/day) | $\delta^2H$ enrichment rate (‰/day) | Ratio of mean enrichment rates | Model-derived isotope fractionation factor $\alpha^{18}O$ | Model-derived isotope fractionation factor $\alpha^2H$ | Ratio of deviations of $\alpha$ from unity |
|---|---|---|---|---|---|---|---|
| toppits | $0.837 \pm 0.016$ | $0.340 \pm 0.021$ | $1.281 \pm 0.069$ | 3.77 | 0.97338 | 0.89753 | 3.85 |
| toppits_double | $0.449 \pm 0.014$ | $0.169 \pm 0.007$ | $0.651 \pm 0.031$ | 3.85 | 0.98787 | 0.94192 | 4.79 |
| Al3z | $0.119 \pm 0.004$ | $0.042 \pm 0.008$ | $0.0959 \pm 0.0353$ | 2.28 | 0.96119 | 0.86295 | 3.53 |
| Al3z_hs | $0.026 \pm 0.002$ | $-0.004 \pm 0.009$ | $0.0002 \pm 0.0428$ | -0.05 | 1.04339 | 1.14501 | 3.34 |

Our calculation of minimum headspace volume (Eq. 3) accounts for $n = 5$ replicates to be safely possible, the analyzer-demanded gas flow rate of $q = 35$ mL/min, and $t = 5$ min usually necessary for reaching a sufficiently long plateau (e.g. 90 s)

in the observed data. Based on these numbers we obtain a volume of 875 mL which we round up to 1 L to have an additional safety margin and for practical reasons. Considering a sample bag with a headspace volume of 1 L and an equilibration temperature of 20°C, we calculated that 17.24 µL of liquid water fully saturate this headspace (Eq. 6 and Eq. 7). At this temperature the isotope separation $\varepsilon$ is about 9.81‰ for $\delta^{18}O$ and 85.21‰ for $\delta^2H$ (Eq. 5). We assume accepted analytical uncertainties of 0.2‰ for $\delta^{18}O$ and 1.0‰ for $\delta^2H$ that should not be exceeded. These result in a minimum water volume of

0.85 mL or 1.47 mL, respectively, which has to be contained in the collected samples (Eq. 4) and be able to exchange with the corresponding headspace during the projected equilibrium time.





## 4 Discussion

### 4.1 Container material

In the first part of this study we investigated ten different bags of various materials and closure types regarding their
capability to hold liquid water and water vapor and found a wide range of weight loss rates spanning three orders of
magnitude. It seems reasonable to assume that all of the weight lost was water. The consistent linearity of the water loss
characteristics observed over the course of several weeks indicates that, generally, export of water vapor was not limited by
the total water content of the field-moist samples. Evaporation from gradually decreasing water reservoirs being the limiting
factor would have resulted in corresponding decreases of water loss rates over time. Instead, water losses were constant and
thus persistent diffusion from well-maintained vapor reservoirs to ambient through bag walls and closures can be deduced.
For identical bag types (toppits, Al3z) different water loss rates were observed in the two parts of the study (1.000 g/(m$^2$ day)
vs. 0.837 g/(m$^2$ day) and 0.155 g/(m$^2$ day) vs. 0.119 g/(m$^2$ day), respectively). We relate this effect to the fact that the two
parts of our study were conducted during different seasons (late winter and summer, respectively) under accordingly varying
humidity conditions in the laboratory where only the temperature was controlled but not the humidity. Notwithstanding the
initial use of a less precise scale, we could clearly show that only few of the tested bags were capable of reliably holding
water vapor inside under ambient temperature conditions (Fig. 2) and are therefore suitable for the DVE-LS method where
the conservative storage of moist soil, rock or plant samples is elementary.

Nonetheless, we further scrutinized not only Al-laminated bags but also transparent freezer bags. Similar to the Al-laminated
bags, they allow for easy handling and have therefore been used previously by our and other research groups (e.g.
Garvelmann et al., 2012). Such transparent freezer bags had been tested for the same purpose before with weight losses of
only 0.06 g in the first ten days (Hendry et al., 2015). However, it is unclear whether those bags and the ones tested here
exactly match in terms of material type and strength. We observed weight differences of up to 10% for seemingly similar
empty freezer bags from different batches (data not shown). We attribute this to potential differences in the production
process resulting in variable wall strengths or some kind of 'age effect' caused by e.g. the outgassing of softeners or material
degradation from UV impacts. Notably, the mean weight loss rate of toppits bags is about twice the mean weight loss rate of
toppits_double bags (0.837 g/(m$^2$ day) vs. 0.449 g/(m$^2$ day), respectively) which confirms that under identical environmental
conditions the water vapor transmission rate is an inverse function of the material thickness. It is therefore quite plausible
that different magnitudes of weight losses and isotope effects for seemingly similar bags have been found under fairly
similar temperature and RH settings in previous studies and in this study.

Theoretical values for water vapor permeability were available for most materials used in this study. However, they had been
determined under different temperature and RH settings presumably given different standardization requirements in different
countries (Tab. 1). We assumed that the permeabilities are not affected by structural changes on the molecular level within
the applied temperature ranges and thus can be normalized to any desired vapor pressure gradient. In doing so, we calculated
water vapor permeabilities for all bags investigated in the first part of our study (data not shown) for the average temperature





and RH conditions recorded in our laboratory. However, comparison of calculated and theoretical permeability values was only somewhat helpful as available values for LDPE did not include material strength and values for Al-laminated bags were only reported as "lower-than" expressions which we consistently undercut. Further, we had assumed that in all cases vapor loss had occurred exclusively through the bags' entire wall areas despite the different tested closure types for Al-laminated bags and some bags lying on their sides or touching other surfaces.

Generally, evaporation is proportional to the saturation vapor pressure deficit expressed in absolute pressure units (e.g. hPa) (Dalton, 1802) which defines the gradient and thus the vapor flux. Under stable temperature conditions it is directly proportional to the saturation vapor pressure deficit, expressed in relative fractions of saturation vapor pressure (1-RH, in %). Having recorded RH in our laboratory, we observed extremes that translate to relative vapor pressure deficits of 88.2% and 16.9% which constitutes a maximum variation factor of 5.2. This factor is still larger than 1.5 when considering only the

mean and standard deviation of RH. These numbers do not fully describe though the variability of individual isotopologues' vapor pressures in the laboratory air. We are convinced that similar conditions hold for other laboratories as well. It is therefore impossible to fully consider all relevant drivers of evaporation and thus isotope effects. However, we argue that this would be necessary when trying to obtain unflawed data from DVE-LS sample bags that fail to prevent significant water loss over the time of the respective isotope assays.

One argument for the use of transparent, yet gas- and thus vapor-permeable sample bags has been their capability to dampen potentially increased concentrations of $CO_2$ caused by ongoing microbial activity in natural soil samples. Also significant levels of spectrally interfering VOCs that perhaps accompany plant water analysis might be levelled out. Yet, the complex field of plant metabolism and related VOC emissions is outside the scope of this study. Changes in the gas matrix have been demonstrated to affect isotope analyses on laser-based analyzers like the one used in this study (Gralher et al., 2016). Unlike

the presence of e.g. alcohols, elevated levels of $CO_2$ are not flagged by the analyzer's data post-processing software ChemCorrect™ (West et al., 2011). However, it has also been described how biases caused by the build-up of $CO_2$ could be reliably corrected with reasonable effort using analyzer-recorded spectral variables only (Gralher et al., 2018). Besides, the fact that biogenic $CO_2$ concentrations may be dampened in the case of gas-permeable bags does not guarantee that they are completely removed and will thus become irrelevant concerning analyzer-immanent gas matrix effects. This relativizes the

presumed advantage of transparent, somewhat gas-permeable freezer bags for the DVE-LS method considering the potential large isotope effects due to water loss and related isotope fractionation.

Furthermore, using vapor-permeable sample bags means accepting a steady loss of water, i.e. a non-zero net vapor flux from the samples to ambient. This also means accepting the fact that by definition no real equilibrium is reached prior to analysis. Instead, temporary steady-state conditions are reached which are variably close to the desired equilibrium. The deviation

therefrom depends on the momentary water loss rate while the duration is additionally a function of absolute sample water content. Both factors are usually unknown and likely variable between samples and relative to co-measured standards. Specifically, the ratio of mean enrichment rates (Tab. 2) and accepted uncertainty should not be taken for the calculation of





generally applicable maximum storage times. The high uncertainties of underlying isotope readings indicate that a sufficient compliance with PIT cannot be assumed.

Throughout the second part of the study, measurements were performed on similar, but not identical bags in order to ensure their structural integrity during the entire storage time prior to isotope analyses. This explains why for triplicates of toppits bags standard deviations for both weight loss and headspace vapor isotope signatures did not steadily increase but varied over time (Fig. 3 and Fig. 4). It proves that water loss characteristics of similar bags can be variable. Using bags with so-revealed structural differences for the preparation of samples and calibration standards will then cause unnoticed violations

of PIT. A correction of this additional error is not possible with reasonable effort, if at all.

Rayleigh-type simulations of isotope evolutions were performed on calibrated liquid water isotope signatures. They yielded fractionation factors that were consistently higher than those reported for kinetic fractionation for both isotope ratios investigated in this study (Gonfiantini, 1986). The deviations from unity of fractionation factors (compare Eq. 5) derived from isotope and weight loss data of Al-laminated bags were higher than those derived from freezer bag data. The narrow

ranges of underlying data in the case of Al-laminated bags caused relatively low parameter sensitivity which can be deduced from the comparatively small changes in RMSE values (Fig. 6) and thus render the respective fractionation factors useless for interpretation despite their better absolute RMSE values. In the case of Al3z_hs the simulation of the quasi-constant weight and isotope data returned inverted model-derived fractionation factors. Clearly, these must be arbitrary artifacts as changes of the respective RMSE values as functions of applied fractionation factors are negligible over the entire range

investigated. Physically – and hypothetically –, inverted fractionation factors would mean that evaporation would release thermal energy instead of consuming it, thereby causing heavier isotopologues to be preferred in this process which clearly contradicts any common (isotope) knowledge. In the case of freezer bags, model-derived fractionation factors displayed much higher sensitivities notwithstanding their somewhat larger minimum RMSE values. This indicates that in this case the applied Rayleigh model adequately represents the physical processes causing the observed changes in isotope readings. We

consider the wider range of underlying isotope and weight loss data of toppits bags to be the reason for the best agreement between the ratio of mean isotope enrichment and the ratio of deviations from unity of model-derived fractionation factors and vice versa for Al3z_hs bags (Tab. 2). Generally, deviations from unity of the fractionation factors were inversely correlated with wall strengths, i.e. diffusional barriers, which in the case of Al3z bags must have consisted mainly of the zip closure. The fractionation factors may thus be plausible but they do not inherit any practical benefit as they should not be

taken for e.g. correction schemes. They can only be taken as proof that water loss via e.g. liquid water dripping can be excluded and instead a combination of isotope fractionating processes, namely evaporation and diffusion, occurred and thus Rayleigh-type evolutions of water stable isotopes appeared. However, they have limited significance as the observed evolutions are still within the quasi-linear parts of typical Rayleigh curves. It should be noted that despite the higher isotope fractionation factors in the case of Al3z bags, the very small overall water loss resulted in comparatively low enrichment of

heavy isotopes on the timescale of our study which must be of premier interest when conducting isotope studies.





### 4.2 Container size

We calculated the minimum headspace volume for DVE-LS sample containers to be ~1 L. This number is based on our analyzer's gas flow demand (~35 mL/min), its response time, and continuous vapor sampling. Provided full inflation, the suggested container size includes a safety margin as it accounts for occasionally necessary prolonged measurement durations
445 e.g. when aiming at specific, consistent standard deviation thresholds for vapor concentrations and isotope readings to be finally reached on the obtained data plateaus. The proposed bag size also enables replicate analyses on identical samples. This would be the case when the readings obtained in the first attempt are doubted for some reason and need to be confirmed. It would also be the case when some or all samples of the respective batch are expected to be affected by build-up of biogenic $CO_2$ and repeated analyses are desired for applying the previously mentioned correction scheme (Gralher et
450 al., 2018).

The bag size needs to be increased when larger-than-usual volumes of sample material are to be collected e.g. in order to account for low water contents (see next section) or when trying to balance unwanted spatial variability. Also, for isotope analyzers with higher gas flow settings it needs to be adapted accordingly. In extreme cases, a design different from the "normal" continuous and linear vapor sampling might be required. An irresolvable mismatch between vapor sample size and
455 analyzer-demanded gas flowrate might call for e.g. circular (e.g. via sample loop, similar to Gaj et al., 2019) or discrete sampling (e.g. via gas-tight syringe). However, such modifications are outside the scope of this study. Notably, light-weight samples are sometimes required for logistical reasons, e.g. when samples are shipped via air-freight or must be physically carried in large numbers through rough terrain by pitiful scientists. In a careful trade-off with the previously described safety precautions (to-be-enabled analysis duration and iterations), smaller sample bags (e.g. 0.5 L) might then be favorable into
460 which smaller-than-usual samples (see next section) can be collected.

### 4.3 Sample size

Regarding the proposed DVE-LS sample size, we agree with previous suggestions (Wassenaar et al., 2008, Hendry et al., 2015) that researchers should not aim at collecting a certain, standard sample volume but instead collect samples containing a minimum volume of water into the bags. Their suggestion of 3 mL was based on observations using double-freezer-bagged
465 samples of various artificially produced moisture contents where samples below 5% gravimetric water content revealed heavy isotope enrichment exceeding the accepted measurement uncertainty. Unfortunately, no weight changes were reported for those samples. Thus, it cannot be fully excluded that the observed variations in isotopic composition were at least in part a result of water vapor loss to ambient. We calculated the minimum necessary absolute water content to be 1.47 mL when using evaporation-safe bags of 1 L headspace volume. In order to obtain accurate isotope data, we strongly suggest that this
470 volume ratio should always be exceeded. This would, however, be violated when filling e.g. previously proposed Reliance[TM] water containers (Mattei et al., 2019) with only 20 mL of water.



Our suggested minimum volume ratio accounts for the fact that depending on the equilibration temperature, a defined amount of sample water will evaporate in order to saturate a given bag's headspace. Using heat-sealed Al-laminated sample bags with proven evaporation-safety (Fig. 2 and Fig. 3), it seems reasonable to assume closed-system conditions once

isotopic equilibria are established. Then, Equation 9 can be applied and solved for $\Delta\delta_{cs}$ in order to calculate the impact of a too-small sample liquid water reservoir on isotope data accuracy. At any given stable temperature the equilibrium isotope separation and the evaporated water volume can be treated as constants. Thus, a larger amount of sample water will lead to a smaller systemic effect on isotope readings and vice versa. We admit that this is probably of minor importance in the case of typical mid-latitude fine-textured soil samples usually containing sufficient water given the dimensions of common soil

coring devices. However, it could easily become relevant in the context of arid and/or coarse-textured soils or compartment-specific sampling of plants, especially when investigating individual specimen. In either case, we recommend in situ analysis of volumetric water content when collecting soil samples. To be on the safe side for either isotope ratio, no less than 2 mL of sample-contained water should be aimed for e.g. in order to account for likely bag and inflation volume uncertainties. This translates to e.g. a minimum of 10 mL of soil with a volumetric water content of 0.2 m³/m³. Having investigated only pure

water samples, we cannot say if this advice is exhaustive also for samples with high solute concentrations or very low volumetric water contents. It has been demonstrated that high concentrations of salt (Horita, 2005 or references therein) or very low moisture contents (Gaj et al., 2019) can have a significant impact on water-vapor isotope equilibrium fractionation. Such effects potentially present in pristine samples are hard to mimic though as would be necessary for the preparation of appropriate calibration standards. Future studies aiming at expanding the applicability of the DVE-LS method may find

appropriate means to correct for these issues.

It should be noted that the impact of this "small-sample" effect on data accuracy depends on the investigated isotopologues. The ratio of typically accepted measurement uncertainty relative to the isotope separation is smaller and therefore less favorable in the case of $\delta^2H$ than for $\delta^{18}O$ (e.g. 1.0‰/85.21‰ vs 0.2‰/9.81‰, respectively @ 20°C). This means that in the case of $\delta^2H$ the evaporation of a fraction of little over 1% will already lead to measurable effects. This fraction is well below

the threshold of 2% which was suggested by Araguás-Araguás et al. (1995) for acceptable kinetic water loss. Further, the effect increases with temperature as the relative increase of saturation vapor pressure (Eq. 8) is stronger than the respective decrease of isotope separation (Eq. 5). Finally, this shift is fully effective when calibration of raw vapor isotope readings is not facilitated by means of similarly affected calibration standards but rather by means of calculated water-vapor isotope equilibrium fractionation (e.g. Majoube, 1971). Nonetheless, even when employing standards for calibration purposes

researchers should not aim at matching the size of (too) small samples and standards as it appears to be impossible to establish fully identical conditions (PIT) in terms of water content, evaporation-effective interfacial area, equilibration time, headspace inflation volume etc.. Rather, they should try to avoid the "small-sample" effect through collection of samples sufficing the suggested volume ratio (< 500:1) of bag headspace and matrix-bound water reservoir.

We did not try to quantify this effect mathematically for the case of vapor-permeable sample containers as we do not

consider them to be closed systems (Fig. 2, Fig. 3) and thus cannot recommend their use anyway. The model-derived isotope





fractionation factors determined for the Toppits® bags (Fig. 5, Tab. 2) are considerably more different from unity than the ones reported for equilibrium fractionation at ambient temperature (Majoube, 1971). Presumably, this translates to even less favorable accepted-uncertainty-to-separation ratios (Eq. 5, Eq. 9). Clearly, even for larger-than-recommended water volumes contained in a sample (here: 5 mL), continuous enrichment of heavy isotopes and thus measurable effects on isotope
readings quickly appear (Fig. 4). For the case of vapor-permeable sample bags, this renders above considerations based on closed-system assumptions obsolete.

## 4.4 Equilibrium time

When DVE-LS samples are left for isothermal equilibration, vapor exchange between a sample's liquid water reservoir and the respective bag's headspace atmosphere will first include only the sample's outermost water "layers". Relative to the
water volume necessary for saturation of the headspace volume (Eq. 6 and 7), this fraction of an entire sample's liquid water reservoir might be small enough for a temporary "small-sample" effect to evolve (see previous section). This results in initial isotope readings to be shifted towards higher values. It is followed by a downshift and the disappearance of the "small-sample" effect due to inward migration of the exchange zones via diffusion in the samples' liquid and vapor phases. We take this as evidence that the determination of sufficiently long DVE-LS equilibration times should not rely solely on the bulk
vapor saturation (representing almost exclusively $H_2^{16}O$) of a sample container's headspace which likely happens within a few hours not only in the case of pure water samples (David et al., 2018; Pratt et al., 2016). We argue that it is impossible to follow the principle of identical treatment (PIT) when applying such short equilibration times. Even if there is a strict consistency of equilibration times between all samples and relative to co-measured calibration standards, there will still be structural differences resulting in different kinetics of all isotopologues (e.g. $H_2^{18}O$, $HD^{16}O$) prior to equilibria that represent
sufficiently large fractions of the samples' and standards' liquid phases (Fig. 4). We therefore suggest that DVE-LS equilibration times for soil samples should be at least two days to allow for sufficiently large representative elementary volume (REV) (Bachmat and Bear, 1987) to evolve. When inward diffusion is impeded e.g. in the case of clayey soil samples, equilibration times should be extended as already pointed out by Wassenaar et al. (2008). But this is safely possible only with evaporation-safe sample containers.

Obviously, the maximum time that should be allowed for isothermal equilibration is limited. In the case of double-layered transparent bags and water volumes of 5 mL we observed isotope enrichment beyond acceptable limits within 2-5 days, depending on the investigated isotope ratio (Fig. 4). Unfortunately, this happens to be the time period suggested for minimum equilibration. Further, this does not consider smaller samples or a given sample's pre-equilibration history during transport and storage. We argue that even collective storage of such samples in coolers (e.g. Wassenaar et al., 2008) or other
confined spaces is not an entirely safe practice. Despite presumed high relative humidity and thus restricted net evaporation in such spaces, isotope exchange between samples would still take place over time via the vapor phase due to heavier isotopologues' individual vapor pressure gradients. This ultimately erases the isotope ratio differences of interest (compare Ingraham and Criss, 1993, 1998). Generally, storage times are not always predictable and thus should be planned with a





buffer due to e.g. unforeseeable instrument failures, restricted analytical capacities or illness of laboratory staff. Prolonged
storage times must also be considered in the case of extensive isotope assays and/or field campaigns in remote areas.
Therefore, we consider transparent bags to be not suitable for DVE-LS analysis.

Over the entire course of our experiments, we did not see significant changes of isotope readings in the case of heat-sealed
Al-laminated bags filled with 5 mL of distilled water. For natural soil samples, however, it has been shown that extensive
equilibration times may lead to e.g. build-up of unwanted, spectrally interfering methane. This must be taken into account for
ongoing microbial activity in samples with high contents of organic carbon from e.g. the uppermost layers of a forest soil.
(Gralher et al., 2018). Fortunately, significant methane build-up can easily be avoided as it occurs only under anoxic
conditions. These can be prevented for quite some time by using well-balanced container sizes (see Eq. 9) and oxygen-
bearing inflation atmospheres (e.g. synthetic air). Then, equilibration times exceeding by far those proposed for clayey
samples (Wassenaar et al., 2008) are safely possible. We suggest that such samples should be somewhat disintegrated inside
the sampling bags in order to increase the exchange-relevant sample surface area. Given the generally low hydraulic
conductivity of clay and the naturally-occurring long-term persistence of fine-scale isotope variations inside such media, we
assume that full equilibration between an entire clayey sample's liquid water reservoir and the sample bag's headspace vapor
is not likely to happen on the timescale of normal DVE-LS assays. Rather, a liquid water fraction (i.e. the REV) as large as
possible being effective and thus avoiding the "small sample" effect can be pursued in this case.

The case of zip-closed-only Al3z bags can be seen as a representation of sample transport and storage as heat-sealing is
generally not applied before inflation. Here, vapor loss must have happened mostly though the zip closure given integer bag
walls being identical to the ones of Al3z_hs bags. Probably due to the small database (n = 3), we were unable to find a
meaningful correlation ($R^2 = 0.15$) between the lengths of zip closures (Tab. 1) and weight loss rates. Nonetheless, we
assume that feasible storage times without significant heavy isotope enrichment inside such samples are considerably longer
than the duration of this study as stored samples are generally kept deflated and rolled up thus restricting potential vapor
diffusion even more.

**5 Conclusion**

We provided empirical evidence as well as physically well-founded considerations that should help users of the direct vapor
equilibration (DVE-LS) method to plan or optimize the parameters of their matrix-bound water isotope sampling and
analysis campaigns. Specifically, we scrutinized the previously controversial aspects, container material and equilibration
time, as well as the volumes of container headspace and sample-contained water including their optimum ratio which had not
been determined before. Regarding sample containers, we convincingly demonstrated the limits of frequently-used
transparent freezer bags, which were strongly contrasted by Al-laminated bags losing virtually no water and ensuring
consistent isotope readings over unprecedentedly long periods when properly heat-sealed. For the first time, Al-laminated
bags allow the applied equilibration time to be adapted exclusively to sample requirements instead of accepting reduced data

quality in a trade-off with material shortcomings as immanent in the case of freezer bags. Regarding the volumes of available container headspace and sample-contained liquid water necessary for precise and accurate analyses we suggest a ratio of no more than 500:1. For absolute numbers of the container headspace volume the analyzer gasflow demand is authoritative. As a standard operation protocol, we recommend users of the DVE-LS method working with isotope analyzers similar to ours to employ heat-sealed Al-laminated sample bags of 1 L volume, to allow for equilibration times of no less than two days and to collect samples containing at least 2 mL of water. We are confident that our findings will help to further strengthen the DVE-LS method's capability of quickly delivering trustworthy and intercomparable isotope data. Moreover, we feel the need to raise awareness for the method's various complex aspects and underlying physical principles that have to be considered in order not to violate the principle of identical treatment. Future efforts should focus on amendments towards better applicability for geologic or organic samples emitting spectrally interfering VOCs. Calibration strategies that fully mimic the effects potentially accompanying natural soil aggregates including extreme conditions are also still missing.

## Data availability

Data are available from the authors upon request.

## Author contributions

BH and BG jointly designed the experiments and performed the data analysis. BH conducted the experiments; BG conceived the theoretical parts and wrote the first draft of the manuscript. BG and BH prepared the manuscript. MW contributed with advice and reviewed the manuscript.

## Acknowledgments

Special thanks go to Ralph Schwab, manager of the local fish counter, who kindly supported this project in the very beginning by happily providing all types of material typically used in his store for fish and meat packaging and lending us in good faith the required heat-sealing device. We would also like to thank the people from Weber Packaging for providing various free samples of their products. BH was supported through the Bio-TGW project (grant no. 02WGW1538B) funded by the German Federal Ministry of Education and Research (BMBF).

## Competing interests

The authors declare that they have no conflict of interest.



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
