# Peer review of "Technical note: Unresolved aspects of the direct vapor equilibration method for stable isotope analysis ( $\delta^{18}O$ , $\delta^{2}H$ ) of matrix-bound water: Unifying protocols through empirical and mathematical scrutiny"

_Hydrology and Earth System Sciences, 2021_

## Referee Comment (RC2)

Review of : Controversial aspects of the direct vapor equilibration method for stable isotope analysis (δ18O, δ2H) of matrix-bound water: Unifying protocols through empirical and mathematical scrutiny
Author(s): Benjamin Gralher et al.
MS No.: hess-2021-255
MS type: Technical note

Reviewer:  Leonard Wassenaar

General Comments

This technical paper provides a very thorough review and assessment of various approaches and materials types aimed at standardizing the isotope analysis of porewaters using the vapor-equilibration and laser analysis approach.  This is a much needed review, and the testing of materials makes this a worthwhile contribution for informing new practitioners about adopting rigorous approaches for porewater isotopic determinations.  I will not repeat the comments of Review #1 – I add here only additional comments to those already made.

Title – I think the word "controversial" is not a good one here, I suggest substituting it with "unresolved".   The former suggests conflict (at least how an English speaker would read it), whereas what you are really trying to address are "unresolved technical challenges".

One clear operational conclusion for me from all the Figures is this:  do not equilibrate longer than 1-2 days, if possible.   This is obviously both a logistical and analytical consideration and dimension, which is a feature of many applied methods.  But this is indeed possible for many types of permeable soils and geological media. Plan the sampling and analyses accordingly.

Para 1 – lines 32.   What is missing here is relaying that DVE-LE is really a proxy approach over the physical extraction of water methods in the past.  We know there are many problems with physical extraction, and so DVE-LS has some advantages, as noted.  (you have this later in the MS – suggest moving it here)

Figure 1 – I suggest replacing "work discipline" in panel 5 with, "time window" of data integration.

Line 62 – I am aware of several student DVE-LS attempts (not published) on plant water and xylem extracts – they have serious issues with a lot of VOC interferences. Suggest removing this sentence for now rather than appearing to promise what is not yet proven.

Line 135  - SOP – define this acronym in the first instance.

Table 1 / line 261 -  I do not see any of the widely used Ziploc® thick-walled double seal freezer bags used here as is suggested in Line 151/Line 261.  The Trademark for Toppits®  is Zipper®, isn't it?  Unless I am mistaken these are separate trademarked brands and products.  These zippers may not be the same between these.  And you do not want to confuse two company separate products! I am aware that Ziploc brand bags are not easily available in Europe. If this was the case for the omission of the Ziploc bags, perhaps note this.   Would be good to add the thickness (in mil).  Its not clear to me what strength means in this context.

Lines 263-264 Costs – would be good to mention Al-bags cost ca. 5.5 times more that plastic so the reader can assess budget implications.

Figure 3 & 4 – one conclusion that seems obvious to me from Figure 3 & 4 is that if you "triple-bagged" Toppits, you would be very close to the performance of the Al bags - is that a reasonable interpretation of the reduction in water loss and isotope effects in these figures? If so, triple bagging would still be 50 % cheaper than Al, albeit a bit more awkward. Can this be tested?

Page 15 – I think you need to need to add a section on implications for Los Gatos laser – the flow rate on ICOS vapor lasers is >700 mL/min. This can be reduced to around 120 ml/min (see original DVE-LS paper), but no lower. This means there are a different set of practical constrains for users of LGR vapor lasers. Sampling frequency is 1Hz for both suppliers.

Line 361 (and discussion). You need to very clearly state that Toppits could be completely different from Ziploc. This section seems to muddy that discussion, leaving a rather unfounded impression that Toppit performance = Ziploc performance. This could be wrong (or correct) until you have data to prove it.

Discussion on lines 380+ – you will recall that the Hendry et al paper suggested cold storage at 100 % RH to avoid the high flux potential of a low RH environment for storage.

Section 4.2 about containers. One foolproof albeit qualitative observational way for detecting leakage is if the bags deflate in <24h after being pressurized with dry air. There is no mention of this, despite its well known that any bag can leak – was this effect observed and used?

Line 458 – remove the word "pitiful"

Line 486 – regarding porewater salinity this reference is also useful since it was done by DEV-LS.
Koehler G, Wassenaar LI, Hendry J. Measurement of stable isotope activities in saline aqueous solutions using optical spectroscopy methods. *Isotopes in environmental and health studies.* 2013;49(3):378-386.

Line 541 – you really mean Toppits brand storage bags are not suitable... isn't that correct? You did not test Ziploc. Do not draw conclusions for products what you did not actually test.

Line 565 – replace controversial with unresolved.

Line 568 - "…. the limits of Toppit brand transparent freezer bags. (again, do not generalize outcomes towards what you did not test). Cautionary notes are of course always welcome, and this paper clearly shows these.

Finally, there seems to be some confirmation bias against using Ziploc brand plastic freezer bags, despite the brand was not tested in this paper, leading to conclusions that can only be verifiably applied specifically to the Toppits brand alone.

Be sure that conclusions are drawn only for products you actually tested and not generalized to those you did not. The findings of this paper, while highly credible, do not fully agree with what others experienced using Ziploc which gives me some hesitation about seeing very strong conclusions towards not using any plastic bags of any kind, and requiring at least 2 days of equilibration. I think this is walking on thin ice without any supporting evidence.

---

## Author Comment (AC1)

**RC1**: 'Comment on hess-2021-255', Anonymous Referee #1, 25 Jun 2021

**Summary**

The authors present a laboratory and modelling study in which they compare different sample container materials and equilibration times for the direct vapor equilibration laser spectrometry (DVE-LS) method. This method was first introduced in 2008 and has since been used in numerous studies, however with inconsistent application of sample containers and equilibration times. Therefore, this manuscript provides a consistent standard operation protocol that will help to enhance data quality and comparability of isotope measurements across future studies that apply the DVE-LS method.

Given the widespread application of the DVE-LS method, the presented study is of great relevance for the research community. Novel data are presented and the conclusions reached are substantiated by the results of the laboratory experiments. The scientific methods are valid and clearly outlined

We thank the referee for the favorable evaluation and thoughtful comments. In the following, the referee comments (in black) are each followed by our response (in blue).

**Mayor comments:**

The laboratory experiments are reasonable and carried-out well. In my opinion, the simulation of Rayleigh-type fractionation is warranted for the less vapor-tight containers; for the very vapor-tight containers (Al3z, Al3z_hs), the simulations yield highly uncertain results because fractionation effects were minimal. I wonder whether the discussion of the simulation results for Al3z and Al3z_hs could be shortened in 3.2 and 4.1 because they don't contribute much to the core message of the paper.

Our intention was a full-scale scrutiny of the DVE-LS method. Therefore we included the mathematical simulation of the observed isotope data. As discussed, the outcome of this effort is not supposed to be used for correction purposes. Rather, we wanted to show that in the case of Toppits bags unfavorable open system conditions exist and cannot be prevented. In the case of Al-laminated bags, however, this issue does not impact feasible storage and equilibration times.

The figures are well-made and informative. Overall, the presentation of the study is well structured and clear, however, the language could partly be improved (some sentences are very long and difficult to understand and I refer to some specific examples below).

I have struggled the most with section 2.2. Although the theories outlined here seem reasonable, I found it difficult to understand the reasoning behind some analysis steps. I would suggest to better structure and explain why and how each analysis step was carried out. For instance:

- The authors write that eq. (1) was used to simulate the theoretical Rayleigh-curve, however, only in Sect. 3.2. it becomes clear that it was used to determine the fractionation factor α. Can this information be included in Sect. 2.2? Was this (fitted) fractionation factor then applied in eq. (5)?

Thanks for alerting us, we will include this information in the updated section 2.2.

Eq. 5 refers to equilibrium conditions (= closed system), whereas Eq. 1 describes the isotope evolution of an open system. Thus, the fitted fractionation factor was not applied in Eq. 5.

- L 209: I'm not familiar with the term "isotope separation" for ε in the context of water stable isotopes studies (Coplen (2011) refers to ε as "isotope fractionation"). Could the authors provide a short mechanistic description of ε and why is a useful parameter for this analysis (especially with respect to the results reported in L 338?

Throughout the manuscript, we replaced "separation" by "enrichment" as we follow the definition of Clark and Fritz (1997) which will be referenced in the revised manuscript.

- What is eq. (9) used for? The explanation "Equation (9) is somewhat similar to equation (4)." Is not very informative.

What we meant to say was that Eq. 9 describes the same physical relationship as Eq. 4.

We will rephrase the respective sentence to: "Equation (9) describes the same physical relationship as equation (4)."

- In eq. (9), how are $V\_H2O,eq$ and $V\_H2O,sam$ determined? Whereas $V\_H2O,eq$ could be measured based on weight differences of the filled containers over time, I don't understand how $V\_H2O,sam$ could be reliably measured.

$V\_H2O,eq$ was calculated (not measured!) using Eq. 6 or Eq. 7.

$V\_H2O,sam$ is the liquid water content of an arbitrary sample. It is also not measured. However, we recommend that a certain threshold volume should be exceeded in order to avoid the 'small sample' effect which is discussed in section 4.3. This threshold volume (now called $V\_H2O,min$) is calculated using Eq. 4.

We will add this explanation to section 2.2 of the revised manuscript.

- L237-240: How did the authors determine the "mean isotope enrichment rates"?

Mean isotope enrichment rates were determined by adding a trend line to the observed data (s. Fig. 4) and calculating the slope thereof.

We will add this explanation to the respective figure captions in the revised manuscript.

- L237: "Ratios of mean isotope enrichment rates were calculated as estimates of the slopes of so-called evaporation lines that water stable isotope data plot on in dual isotope space when affected by gradual evaporitic enrichment of heavy isotopes. We compared these to the ratio of deviations from unity of the model-derived isotope fractionation factors α (Eq. 1)." What will this comparison analysis tell us?

The 'ratio of mean enrichment rates' as well as the 'ratio of deviations from unity' are two ways to calculate the slope of an evaporation line in dual isotope space (the numbers can be found in Tab. 2). Comparison of the two numbers is a way to check their plausibility. We rephrased the respective paragraph in the discussion section to:

"The wider ranges of underlying isotope and weight loss data of toppits bags are the reason for the higher respective parameter sensitivity. Especially in this case, we consider the good agreement between the ratio of mean isotope enrichment and the ratio of deviations from unity of model-derived fractionation factors (Tab. 2) to be proof for their plausibility."

- L240: "Individually, these deviations yield the respective isotope separations (Eq. 5)." I think that this sentence is relevant for eq. (5) and should therefore be moved there.

This sentence really is about the aforementioned deviations, not the separations (or rather enrichments). The reference to Eq. 5 was provided for additional guidance of the reader.

**Minor comments:**

L42-44: It is not clear what "manifested" and "enabled" refer to. Can the authors rephrase the sentence to be more specific?

We regret the confusion. We will reformat the respective sentence to:

"The growing distribution of laser-based water stable isotope analyzers in recent years and the DVE-LS method's relative simplicity, resulting from fairly little sample preparation workload, low-cost consumables and omission of sophisticated water extraction lines and analyzer peripherals, enabled its rapid, wide-spread adoption."

L75-78: I would suggest to use commas in this sentence or make two sentences out of it.

Commas will be added as suggested.

L81: What is meant by "measurement iterations"?

Details of this post-correction concept can be found in the provided reference. For better readability, we will rearrange the respective sentence to:

"They also presented a post-correction scheme of potentially affected DVE-LS samples based on an analyzer-recorded spectral variable and measurement iterations (Gralher et al., 2018)."

L105: I don't understand the example "groundwater vs. root uptake water". Do the authors mean that a study focusing on groundwater samples will need to adapt a different correction strategy than a study focusing on root uptake water? Why?

This example was given in the aforementioned study of Wang et al. 2019. We will rephrase the respective sentence to make this clearer.

L128-131: I would suggest to use commas in this sentence or make two sentences out of it.

Comma will be added as suggested.

L173: This sentence seems to be out of context. Can the authors explain why "each of these bag candidates were then equipped on one side with custom-made septa of silicone blots or adhesive tape"?

We will rephrase the sentence to:

"In total, 21 replicates of each of these bag candidates were then equipped on one side with silicone blots or adhesive tape which served as custom-made septa during direct headspace analyses."

L185: Do the authors mean the Al3z_hs bags? Also, it is not clear to me how the PIT was implemented if the same bag type was used as reference for the other (different) bag types.

Yes, we mean the Al3z_hs bags.

For clarification we will replace "otherwise" by "apart from that".

L175, L187: Were the sample water and the standard water isotopically identical? How many standards were used?

Sample water and standard water were not isotopically identical. On each day of analysis two standards were co-measured.

We will add this information to the manuscript.

L338: If the authors used eq. (5) to calculate ε, what are the respective fractionation factors?

Temperature-dependent isotope fractionation factors α were calculated using the pertinent equation provided by Majoube (1971).

We will include the respective numbers in the manuscript.

L446: How would it be possible to re-measure a punctured bag again or should punctured bags be discarded regardless? I could imagine that after the measurement is done, the needle puncture could be sealed again with silicone blots or adhesive tape.

Pre-applied silicone blots provide sufficiently closed septa allowing for re-measurements. This has been done before (see e.g. Gralher et al., 2018). However, for this study we intended to perform only singular measurements on individual bags in order to ensure their structural integrity during the entire storage time prior to isotope analyses. In doing so, we wanted to ensure identical conditions for all investigated bags.

From our experience, the best way to allow re-measurements is to apply to each bag the respective number of silicone blots well in advance (~ 2 days). Freshly applied silicone outgases VOCs which likely flaws isotope readings.

We added the following sentence to the method section:

"In order to account for outgassing of VOCs from freshly applied silicone and thus compromising isotope readings, this step was conducted well in advance (≥2 days) of the isotope analyses."

L452: What is meant by "…trying to balance unwanted spatial variability."?

This is a hypothetic scenario, where researchers might want to collect larger than usual samples. Theoretically this could be applied in cases of presumably high soil heterogeneity by mixing adjacent samples from identical soil depths into one sampling bag.

L468: Could the authors include a recommendation for the required minimum volume of liquid isotope standards used for calibration and drift control?

For clarification we will add the following sentence to section 4.3:

"For liquid water standards, prepared for calibration and drift control purposes of DVE-LS samples, the same holds true. This means that they also need to consist of at least this water volume."

L475: "Then, Equation 9 can be applied and solved for delta-delta_CS in order to calculate the impact of a too-small sample liquid water reservoir on isotope data accuracy." Can the authors be more specific? How would $\varepsilon$, $V\_H2O,eq$ and $V\_H2O,sam$ or $f$ be determined?

The definitions of the listed parameters are provided in section 2.2.

L504: Does "this effect" refer to "the small-sample effect"?

Yes, it does.

L555: "The case of zip-closed-only Al3z bags can be seen as a representation of sample transport and storage…" In introductory sentence will make it easier to grasp the motivation of this paragraph. A suggestion:

The comparison of zip-closed-only Al3z bags and heat-sealed zip-closed Al3z_hs bags allow us to assess the negative impacts of sample transport and storage on vapor loss.

We will rephrase the beginning of the respective paragraph to:

"The case of zip-closed-only Al3z bags can be seen as a representation of sample transport and storage. Comparison of zip-closed-only Al3z bags and heat-sealed zip-closed Al3z_hs bags allows assessing the negative impacts of sample transport and storage on vapor loss, as heat-sealing is generally not applied before inflation."

L560: Could the authors provide some more detailed recommendations on how to store filled sample containers if isotope analysis is possible only after more than 2 days? Should the containers be stored in a fridge/freezer until analysis?

We will add the following recommendation to the manuscript's conclusion section.

"In order to prevent evaporation, Al-laminated bags do not require extra measures. Nonetheless, cooling samples prior to inflation is advisable in order to reduce microbial activity as well as the associated build-up of $CO_2$ and changes of the gas matrix. Ultimately, this prevents reducing environments and the production of spectrally interfering gases. Freezing samples for this purpose, however, cannot be recommended as this might destroy soil aggregates and microstructures. The resulting effect on isotope readings has not yet been investigated systematically."

L569: "…over unprecedentedly long periods …". Can the authors be more specific here?

In this conclusional sentence we refer to the fact that samples stored in the proposed Al-laminated bags did not lose significant amounts of water over a time period of four weeks, which exceeds any routine storage time we are aware of.

L580: What is meant by "extreme conditions"?

We refer to "extreme conditions" such as cases of high salinity or strong aridity.

We will insert "regarding e.g. salinity or aridity" after "extreme conditions".

References

Coplen, T. B.: Guidelines and recommended terms for expression of stable-isotope-ratio and gas-ratio measurement results, Rapid Commun Mass Sp, 25, 2538–2560, https://doi.org/10.1002/rcm.5129, 2011.

**Citation**: https://doi.org/10.5194/hess-2021-255-RC1

Clark, I. D. and Fritz, P.: Environmental Isotopes in Hydrogeology, Lewis, Boca Raton. 348 p., doi: 10.1201/9781482242911, 1997.

Gralher, B., Herbstritt, B., Weiler, M., Wassenaar, L.I., and Stumpp, C.: Correcting for biogenic gas matrix effects on laser-based pore water-vapor stable isotope measurements, Vadose Zone J. 17:170157, doi: 10.2136/vzj2017.08.0157, 2018

Majoube M.F.: Fractionnement en oxygène−18 et en deuterium entre l'eau et sa vapeur, J. Chem. Phys. 58:1423–1436, 1971.

Wang, H., Si, B., Pratt, D., Li, H., Ma, X.: Calibration method affects the measured $\delta 2H$ and $\delta 18O$ in soil water by direct $H_2Oliquid$–$H_2Ovapour$ equilibration with laser spectroscopy, Hydrol. Process., 9, 1–11, doi:10.1002/hyp.13606, 2019.

---

## Author Comment (AC2)

Review of: Controversial aspects of the direct vapor equilibration method for stable isotope analysis (δ18O, δ2H) of matrix-bound water: Unifying protocols through empirical and mathematical scrutiny
Author(s): Benjamin Gralher et al.
MS No.: hess-2021-255
MS type: Technical note

Reviewer: Leonard Wassenaar

General Comments

This technical paper provides a very thorough review and assessment of various approaches and materials types aimed at standardizing the isotope analysis of porewaters using the vapor-equilibration and laser analysis approach. This is a much needed review, and the testing of materials makes this a worthwhile contribution for informing new practitioners about adopting rigorous approaches for porewater isotopic determinations. I will not repeat the comments of Review #1 – I add here only additional comments to those already made.

We thank the referee for the favorable evaluation and thoughtful comments. In the following, the referee comments (in black) are each followed by our response (in blue).

Title – I think the word "controversial" is not a good one here, I suggest substituting it with "unresolved". The former suggests conflict (at least how an English speaker would read it), whereas what you are really trying to address are "unresolved technical challenges".

We will change the title to:
"Unresolved aspects of the direct vapor equilibration method for stable isotope analysis (δ18O, δ2H) of matrix-bound water: Standardizing protocols through empirical and mathematical scrutiny"
to emphasize the importance of the scrutinized aspects as well as the fact that a consensus regarding these had not been reached yet within the community.

One clear operational conclusion for me from all the Figures is this: do not equilibrate longer than 1-2 days, if possible. This is obviously both a logistical and analytical consideration and dimension, which is a feature of many applied methods. But this is indeed possible for many types of permeable soils and geological media. Plan the sampling and analyses accordingly.
We agree that in an ideal setting, large numbers of samples can be processed within a few days. However, wouldn't it be nice to have the option to plan with a buffer that accounts for potential issues regarding cooling chain, transport time, remote field sites, unknown soil types etc.? The suggested Al-laminated bags would provide this option.

Para 1 – lines 32. What is missing here is relaying that DVE-LE is really a proxy approach over the physical extraction of water methods in the past. We know there are many problems with physical extraction, and so DVE-LS has some advantages, as noted. (you have this later in the MS – suggest moving it here)
We will rephrase the respective sentence to:
"Instead of physically extracting water, the method employs analysis of a corresponding vapor phase and thereby bypasses many of the previously necessary, laborious sample preparation steps."

Figure 1 – I suggest replacing "work discipline" in panel 5 with, "time window" of data integration.

We will add "time window" to the list of critical aspects. "Work discipline" refers to the higher number of samples that can be processed if the measurements are conducted in a structured and disciplined manner.

Line 62 – I am aware of several student DVE-LS attempts (not published) on plant water and xylem extracts – they have serious issues with a lot of VOC interferences. Suggest removing this sentence for now rather than appearing to promise what is not yet proven.
True, we ourselves have also tried DVE-LS analyses of plant and xylem samples and encountered related VOC issues. We will insert "published" before "field study".

Line 135 - SOP – define this acronym in the first instance.
Done

Table 1 / line 261 - I do not see any of the widely used Ziploc® thick-walled double seal freezer bags used here as is suggested in Line 151/Line 261. The Trademark for Toppits® is Zipper®, isn't it? Unless I am mistaken these are separate trademarked brands and products. These zippers may not be the same between these. And you do not want to confuse two company separate products! I am aware that Ziploc brand bags are not easily available in Europe. If this was the case for the omission of the Ziploc bags, perhaps note this.
Thank you for raising this point. Actually, 'Ziploc' brand was in fact used by Toppits for their freezer bags at the time when we conducted the weight loss experiments. As proof, we provide here a picture of an originally used Toppits freezer bag as well as the package in which they were sold.

[Figure]

[Figure]

Apparently, in the meantime Toppits changed the zip closure from 'Ziploc' to 'Safeloc' brand, which may have caused the confusion. However, 'Ziploc' and 'Safeloc' branded Toppits freezer bags are

identical in terms of material type and thickness. This should make our findings transferable to nowadays available 'Safeloc' bags.

To avoid confusion to the readers, we will remove the word 'Ziploc' in line 261. Except for line 261, the bags were called 'Toppits' throughout the manuscript.

Overall, we would like to emphasize that the distinction should not be made between 'Toppits' and 'Ziploc' but rather between 'Toppits' versus e.g. 'SC Johnson', regarding the manufacturer/distributor potentially using different material types and thicknesses, or 'Ziploc' versus e.g. 'Safeloc' brand which refers to the closure type.

Would be good to add the thickness (in mil). Its not clear to me what strength means in this context.
Thickness data in micrometers (µm) are provided in Table 1. The header will be changed from "material strength" to "material thickness"

Lines 263-264 Costs – would be good to mention Al-bags cost ca. 5.5 times more that plastic so the reader can assess budget implications.
At this point, we meant to emphasize that both tested bags are available at low costs, ranging two orders of magnitude below e.g. specialty Tedlar bags or Linde plastigas® bags, which cost about € 25/unit and are thus prohibitive.

Figure 3 & 4 – one conclusion that seems obvious to me from Figure 3 & 4 is that if you "triple-bagged" Toppits, you would be very close to the performance of the Al bags - is that a reasonable interpretation of the reduction in water loss and isotope effects in these figures? If so, triple bagging would still be 50 % cheaper than Al, albeit a bit more awkward. Can this be tested?
We agree that, theoretically, triple bagging samples using freezer bags (€ 0.42/"unit" (a.k.a sample)) are ca. 36% cheaper than Al-laminated bags (€ 0.65/unit).  Further, it can be assumed that water loss rates and isotope effects would be about one third of the effects observed for standalone Toppits bags due to the threefold material thickness. However, we hesitate to say that the effects would be "very close" to the case of Al-laminated bags. Instead, the latter – even without heat-sealing – would still perform 2.3, 2.7, and 4.4 times better regarding water loss rates, $\delta^{18}$O and $\delta^{2}$H. Apart from that, we would like to emphasize that in our lab routine we never use inflated Al bags without heat-sealing. We always work with heat-sealed bags, which outperform the theoretically triple bagged samples by factors of 10.7, 28.3, and 2135(!) for water loss rate, $\delta^{18}$O and $\delta^{2}$H.
Further, from our experience, double-bagging bears the risk that the zip closure of the inner bag pops open during or after inflation when inserted into another bag of the same size. Clearly, this risk would increase when tripling the bags. Not to mention the additional time needed for handling of so many bags while dealing with evaporation-susceptible samples. Using freezer bags of different sizes might make smoother work but also increases the relevant areas of vapor-permeable materials, the effect of which would have to be quantified in a different study.

Page 15 – I think you need to add a section on implications for Los Gatos laser – the flow rate on ICOS vapor lasers is >700 mL/min. This can be reduced to around 120 ml/min (see original DVE-LS paper), but no lower. This means there are a different set of practical constrains for users of LGR vapor lasers. Sampling frequency is 1Hz for both suppliers.
This issue has already been addressed in the Discussion (l. 452) where we state that for users employing analyzers with higher gas flow demands larger sample bags or a different sampling design are necessary. We will add "(e.g. Los Gatos)" to the revised manuscript.

Line 361 (and discussion). You need to very clearly state that Toppits could be completely different from Ziploc. This section seems to muddy that discussion, leaving a rather unfounded impression that Toppit performance = Ziploc performance. This could be wrong (or correct) until you have data to prove it.

As elaborated above, we used 'Toppits' bags in our study, where the closure type was 'Ziploc'. Certainly, this should not be mixed up with 'Ziploc'-closed bags from a different manufacturer (e.g. SC Johnson?), apparently supplying other continents and called "Ziploc bags" in other studies.
To make it clearer, we will change the sentence
"However, it is unclear whether those bags and the ones tested here exactly match in terms of material type and strength."
to
"However, it is unclear to what extent those bags and the ones tested here match in terms of material type and strength as we only tested standard freezer bags available in supermarkets and drugstores in Germany"

Discussion on lines 380+ – you will recall that the Hendry et al paper suggested cold storage at 100 % RH to avoid the high flux potential of a low RH environment for storage.
We are aware of the suggestions of the Hendry et al paper.
However, Ingraham and Criss (1991, 1998) demonstrated that 100% RH does not prevent shifts in isotopes of adjacent liquid water reservoirs due to relative deviations from equilibrium regarding the heavier isotopologues (different vapor pressure deficits of the isotopologues).

Section 4.2 about containers. One foolproof albeit qualitative observational way for detecting leakage is if the bags deflate in <24h after being pressurized with dry air. There is no mention of this, despite its well known that any bag can leak – was this effect observed and used?
We are fully aware of this qualitative test. However, having worked with Al-laminated bags for several years, we literally never saw a leaking bag when properly heat-sealed - and handled, of course.

Line 458 – remove the word "pitiful"
Will be removed as suggested.

Line 486 – regarding porewater salinity this reference is also useful since it was done by DEV-LS.
Koehler G, Wassenaar LI, Hendry J. Measurement of stable isotope activities in saline aqueous solutions using optical spectroscopy methods. *Isotopes in environmental and health studies.* 2013;49(3):378-386.
Thank you for alerting us to this reference, which will be inserted into the updated manuscript.

Line 541 – you really mean Toppits brand storage bags are not suitable… isn't that correct? You did not test Ziploc. Do not draw conclusions for products what you did not actually test.
As elaborated above, we used 'Toppits' bags in our study, where the closure type was 'Ziploc'. Certainly, this should not be mixed up with 'Ziploc'-closed bags from a different manufacturer (e.g. SC Johnson?), apparently supplying other continents and called "Ziploc bags" in other studies.

Line 565 – replace controversial with unresolved.
Will be changed as suggested.

Line 568 - "…. the limits of Toppit brand transparent freezer bags. (again, do not generalize outcomes towards what you did not test). Cautionary notes are of course always welcome, and this paper clearly shows these.
We will insert "Toppits" as suggested.

Finally, there seems to be some confirmation bias against using Ziploc brand plastic freezer bags, despite the brand was not tested in this paper, leading to conclusions that can only be verifiably applied specifically to the Toppits brand alone.
Be sure that conclusions are drawn only for products you actually tested and not generalized to those you did not. The findings of this paper, while highly credible, do not fully agree with what others experienced using Ziploc which gives me some hesitation about seeing very strong conclusions

towards not using any plastic bags of any kind, and requiring at least 2 days of equilibration. I think this is walking on thin ice without any supporting evidence.

As elaborated above, we used 'Toppits' bags in our study, where the closure type was 'Ziploc'. Certainly, this should not be mixed up with 'Ziploc'-closed bags from a different manufacturer (e.g. SC Johnson?), apparently supplying other continents and called 'Ziploc bags' in other studies.

Apart from that, we certainly never intended to defame the work of other research groups – especially not the work that was inspiration for this study.

References

Ingraham, N. L. and Criss, R. E.: Effects of surface area and volume on the rate of isotopic exchange between water and water vapor, J. Geophys. Res.-Atmos., 98, 20547–20553, doi:10.1029/93jd01735, 1993.

Ingraham, N. L. and Criss, R. E.: The effect of vapor pressure on the rate of isotopic exchange between water and water vapor, Chem. Geol., 150, 287–292, doi:10.1016/s0009-2541(98)00109-0, 1998.